# Using the Aqueous Phase Produced from Hydrothermal Carbonization Process of Brown Seaweed to Improve the Growth of *Phaseolus vulgaris*

**DOI:** 10.3390/plants12142745

**Published:** 2023-07-24

**Authors:** Damiano Spagnuolo, Viviana Bressi, Maria Teresa Chiofalo, Marina Morabito, Claudia Espro, Giuseppa Genovese, Daniela Iannazzo, Patrizia Trifilò

**Affiliations:** 1Department of Chemical, Biological, Pharmaceutical and Environmental Sciences, University of Messina, Viale Ferdinando Stagno D’Alcontres 31, 98166 Messina, Italy; chfmtr98h57f158j@studenti.unime.it (M.T.C.); morabitom@unime.it (M.M.); ggenovese@unime.it (G.G.); ptrifilo@unime.it (P.T.); 2Department of Engineering, University of Messina, Contrada di Dio, Vill. S. Agata, 98166 Messina, Italy; vibressi@unime.it (V.B.); espro@unime.it (C.E.); daniela.iannazzo@unime.it (D.I.)

**Keywords:** circular economy, gas exchange, HTC, plant productivity, seaweed biomass, seed priming

## Abstract

Seaweeds are considered a biomass for third-generation biofuel, and hydrothermal carbonization (HTC) is a valuable process for efficiently disposing of the excess of macroalgae biomass for conversion into multiple value-added products. However, the HTC process produces a liquid phase to be disposed of. The present study aims to investigate the effects of seed-priming treatment with three HTC-discarded liquid phases (namely AHL_180_, AHL_240_, and AHL_300_), obtained from different experimental procedures, on seed germination and plant growth and productivity of *Phaseolus vulgaris* L. To disentangle the osmotic effects from the use of AHL, isotonic solutions of polyethylene glycol (PEG) 6000 have also been tested. Seed germination was not affected by AHL seed-priming treatment. In contrast, PEG-treated samples showed significantly lower seed germination success. AHL-treated samples showed changes in plant biomass: higher shoot biomass was recorded especially in AHL_180_ samples. Conversely, AHL_240_ and AHL_300_ samples showed higher root biomass. The higher plant biomass values recorded in AHL-treated samples were the consequence of higher values of photosynthesis rate and water use efficiency, which, in turn, were related to higher stomatal density. Recorded data strongly support the hypothesis of the AHL solution reuse in agriculture in the framework of resource management and circular green economy.

## 1. Introduction

In the last twenty years, climate change and the associated increase in the frequency of extreme climate events have severely affected agricultural productivity. Based on the most recent IPCC report, during the years 1983–2009, about three quarters of the global harvested areas experienced yield losses induced by drought events, and further negative impacts on crop yields are expected in the future [1]. The socio-economic effects of these projections will be dramatically emphasized by the forecasted increase in population growth and the resulting food requirement [2,3]. Consequently, one of the most urgent challenges of researchers is to find solutions that limit or, at least, mitigate the climate-change-driven loss in plant productivity.

Macroalgae have been used as a plant fertilizer and/or biostimulant since ancient times. Nevertheless, the increasing number of pieces of experimental evidence, mainly recorded in the last two decades, has laid the scientific foundations on the benefits of using seaweed extracts to improve plant productivity, particularly under environmental constraints [4,5,6,7,8,9,10]. Finally, besides the advantage of being a low-cost fertilizer and growth biostimulant [11], macroalgal extracts also offer the advantage of valorizing the excess of seaweed biomass that occurs in eutrophic environments. In this light, seaweeds can actually represent a key product for achieving eco-sustainable agriculture solution.

Recently, macroalgae have been used also as a biomass to produce carbon-rich solid fuel by a thermochemical process called hydrothermal carbonization (HTC) [12,13,14,15,16].

HTC is a process converting biomass into a coal-like product. In the past, it has been used to simulate natural coalification in the laboratory. However, in the last years, HTC has been reconsidered as an alternative to transform wet biomass into a value-added product, including biofuel [17,18]. Unlike conventional dry thermochemical processes (e.g., combustion, pyrolysis, and gasification), HTC does not require an expensive or energy-intensive preliminary drying step, as it directly exploits the water retained in digestate as a solvent during the process [19]. In particular, the HTC process takes place between 180 °C and 300 °C and at high pressure (i.e., between 10 bar and 80 bar), with residence time ranging from a few minutes to several hours. The HTC process converts wet feedstock into a carbonaceous solid fraction, known as hydrochar, and a liquid phase named aqueous HTC liquid (AHL); small quantities of gas can be also produced [20,21]. The AHL phase is commonly discarded at the end of the process. However, it is still rich in minerals, organic compounds [22,23], and, in some cases, heavy metals and toxic compounds [24]. Therefore, the disposal of aqueous HTC solutions is one of the major limits of using HTC. However, its richness in nutrients and compounds encourages studying possible applications.

In the present study, we investigated the effects of the discarded AHL solutions obtained by HTC of seaweed feedstock on plant productivity and water relations of *Phaseolus vulgaris* L. cv. ‘Borlotto’ plants, a worldwide relevant food crop. In detail, we performed bean-seed-priming treatment with three AHL solutions produced by different experimental protocols of the HTC process of *Sargassum muticum* (*Phaeophyceae*) biomass in order to investigate the potential reuse of the discarded aqueous HTC liquid, with advantages for crop productivity. To the best of our knowledge, few studies have investigated the reuse of water process obtained from HCT to improve crop yields [23,25], and no investigation on its potential reuse for seed priming has been performed.

Seed priming is an agricultural practice that consists of pre-germination seed imbibition in solutions containing chemical or biological priming agents. In fact, the osmotic stimulus and/or the presence of specific molecules in the solutions used for the controlled seed hydration elicits the activation of metabolic and genetic pathways. This, in turn, can improve not only the germination success but also plant productivity and even seedling resistance to biotic and abiotic stress [26,27,28,29,30].

On this basis, this study aimed to (i) investigate the potential benefits of a seed-priming treatment using AHL obtained by macroalgae HTC feedstock and (ii) disentangle the potential benefit(s) of seed hydration using the discarded aqueous HTC liquids from that of a mere osmotic effect. To achieve this goal, a set of seeds was imbibed with polyethylene glycol (PEG) solutions with an osmotic pressure similar to that of the AHL solutions (Figure 1). We expected that seed treatment with discarded aqueous HTC liquids may improve plant performance, offering an added value to the yet explored valorization of seaweeds by HCT [13] as well as to the possible use of macroalgae in several industrial activities [31].

## 2. Results

Seeds germination was about 80% under all tested conditions except that in samples treated with PEG solutions, in which statistically lower seed germination occurred (i.e., about 65%; Figure 2).

The three AHL solutions differently affected the plant biomass of bean samples. All samples showed a similar number of leaves per plant (i.e., about 15). However, a statistically higher whole leaf surface area was recorded in samples treated with AHL_180_ solution because of a higher mean leaf surface area (Figure 3). It can be noted that also AHL_240_, AHL_300_, and PEG_AHL300_ samples showed a tendentially higher mean leaf surface area with respect to control plants. In AHL_300_ and PEG_AHL300_ samples, the highest leaf mass area (LMA) and the lowest leaf dry matter content (LDMC) values were recorded. 

AHL-treated samples showed changes in the whole plant biomass and, in some samples, changes in the shoot/root ratio with respect to the values recorded in control plants (Figure 4). In detail, AHL_180_ solution treatment promoted an increase in the stem length, stem dry weight, and whole leaf dry weight with respect to C samples. This, in turn, led to a higher whole plant biomass and, mainly, higher shoot/root ratio values in AHL_180_ vs. C samples. At the time of measurements, AHL_180_-treated samples also showed a tendentially higher number of fruits per plant with respect to the control (Appendix A). A shoot/root ratio value as high as about 5 and statistically similar to that recorded in AHL_180_ samples was recorded in AHL_300_ and PEG_AHL300_ samples (Figure 4). It can be noted that the highest root dry weight values were recorded in plants treated with AHL_240_ as well as in samples treated with a similar osmotic pressure solution (i.e., PEG_AHL240_ samples). 

No differences in the turgor loss point and osmotic potential at full turgor were recorded among the measured set of plants (Figure 5). By contrast, samples treated with AHL solutions showed higher stomatal conductance to water vapor, photosynthesis rate, water use efficiency (Figure 5), as well as higher stomatal density (Table 1) with respect to control plants. A lower leaf C:N ratio, due to the increase in the leaf nitrogen content, occurred in AHL-treated samples with respect to control samples (Table 1). 

Moreover, the spectra of the qualitative X-ray microanalyses of the leaves showed peaks of Cu, Na, and Zn in AHL-treated samples but not in control leaves (Figure 6). In this regard, however, it should be noted that the technique of the scanning electron microscopy and energy-dispersive X-ray spectroscopy produces qualitative rather than quantitative data. Stomatal density affected gas exchange and water use efficiency, and these, in turn, were positively related to the shoot biomass (i.e., leaf and stem) (Appendix A).

## 3. Discussion

Present findings support the hypothesis that AHL obtained using *Sargassum muticum* (*Phaeophyceae*) as HTC feedstock can be a valuable resource in agriculture practices. In fact, our data clearly showed that even if seed-priming treatment with the AHL did not affect seed germination, AHL_180_ treatment improved the growth of bean plants under well-watered conditions. The results recorded in PEG-treated samples allowed us to conclude that this result was not due to a mere effect of the osmotic pressure of the wastewater. On this basis, the present study is a starting point for future research in which the composition of the used AHL is characterized from a chemical point of view to understand which component or mixture of compounds affects plant performance. Moreover, it is interesting to investigate whether, in addition to growth improvement, AHL seed-treated samples are also more resistant to environmental constraints.

### 3.1. Effects of the Seed-Priming Treatment with AHL Solutions on Seed Germination

To the best of our knowledge, despite its potential, few studies investigated the reuse of water process obtained from HCT to improve crop yields. Among these, different research works focused on the toxicity of AHL on seed germination. In some of these studies, AHL was mixed with other substrates as the hydrochar itself [32,33,34,35]. Most recently, Celletti et al. [23] tested the effects on the growth of a whole *Zea mays* plant using diluted AHL solutions as a hydroponics medium. The authors recorded no phytotoxic effects on maize. However, the used diluted growth solutions caused some deficiency symptoms. More promising results were recorded by testing AHL solutions obtained by micro- and macroalgal HTC feedstock. Levine et al. [36] reported that the aqueous phase obtained by hydrothermal carbonization algal feedstock supported the growth of microalgae better than a medium containing only inorganic nutrients. Overall, literature data indicate that the conflicting and species-specific results [25,37] are likely caused by the different feedstock used and/or HTC protocols [38]. As far as we know, the study by Wang et al. [39] is the only investigation aimed to check the effects of AHL obtained by macroalgal feedstock on crops. Their results showed that the HTC aqueous solutions, obtained by avoiding the recirculate water process, did not inhibit plant germination. 

In our study, we did not recycle the water process and recorded on *Phaseolus vulgaris* L. cv. ‘Borlotto’ seeds the same effects reported by Wang et al. [25] on cabbage seeds. In other words, even using the HTC feedstock of different seaweeds (i.e., *Laminaria* versus *Sargassum*) and testing the corresponding AHL on different plant species (bean versus cabbage), the reuse of non-recirculated water process to improve plant growth is very promising because it does not affect seed germination and stimulates plant productivity (see comment below). From this point of view, the conversion of the seaweed biomass into synthetic coal and carbon material for liquid contaminant adsorption by HTC [40] is effectively an evaluable carbon-efficient resource use toward improving circular economy in future application. 

It can be noted that a lower germination success was recorded in PEG-treated seeds. Polyethylene glycol is a high-molecular-weight compound, water soluble, and unable to pass through the cell wall. For these reasons, it is commonly used to induce osmotic stress. Despite its relatively wide use, however, some studies reported that the effects induced by PEG solutions are often more consistent compared with those induced by other osmotic compounds, even with the concentrations being equal [41,42]. The higher adverse effects of PEG treatment may be explained by possible contaminants in PEG solutions [43]. On the other hand, different results on seed germination success are obtained as a function of the PEG concentration used and the measured plant species [44,45,46]. Considering literature data and results recorded in the present study, the use of PEG remains one of the reliable approaches to study osmotic effects and/or disentangle, in vivo as well as in vitro, the effects of the treatment with other organic and inorganic compounds from the effects of osmotic stress [47,48,49,50]. In fact, except for a lower germination success, our PEG-treated samples showed the same behavior recorded in control samples and/or in the corresponding iso-osmotic AHL-treated samples. In other words, our experimental plan allowed effectively disentangling the physiological effects of seed priming as driven by mere osmotic stress from that caused by AHL treatment. 

### 3.2. Effects of the Seed Priming with AHL Solutions on Plant Growth

The three studied AHL solutions did not have a similar impact on *P. vulgaris* cv. ‘Borlotto’ growth. Seed-priming treatment with all three solutions led to higher photosynthesis rate and WUE, probably as a consequence of the higher stomatal density [51] and micronutrient content [52] recorded in AHL-treated samples compared with values recorded in control plants. However, the biomass was allocated differently among the three AHL-treated plants. In fact, higher shoot biomass was recorded especially in AHL_180_ samples. By contrast, AHL_240_ and AHL_300_ samples showed higher root biomass. The differences in plant yield and shoot biomass allocation were clearly caused by the use of AHL solution and not by the mere effect of osmotic treatment, at least in AHL_180_ samples. In fact, PEG_AHL180_-treated samples showed gas exchange, stomatal density, and a shoot/root ratio similar to control samples. Conversely, the increase in root biomass recorded in AHL_240_ samples was probably caused by an osmotic stimulus because it was also recorded in PEG_AHL240_ samples. AHL_300_-treated samples showed higher photosynthesis rate and WUE, higher LMA values (as also recorded in all PEG-treated samples), and a biomass of stem and root tendentially higher than that of the control samples. 

Higher root biomass and higher LMA values are physiological traits commonly recorded in plants experiencing drought events. Root traits play a key role in the plant ability to successfully deal with drought events [53,54,55,56,57], and shifting the root-to-shoot ratio can improve the plant water uptake [58]. In fact, in response to drought, a significant increase in the root-to-shoot mass ratio was recorded in more than 120 published studies [59]. Similarly, higher leaf N content (as recorded especially in AHL_180_- and AHL_300_-treated samples) and LMA values are coupled to species living in more xeric environments [60] because an indirect link may exist between LMA and drought resistance [61]. In other words, our results strongly encourage studying the possible benefits of AHL_180_ and AHL_300_ seed-priming treatments to improve plant productivity and/or to stimulate plant tolerance in conditions of low water availability.

## 4. Materials and Methods

### 4.1. Obtaining AHL Solutions

Five grams of air-dried barcoded (BOLD: BRAPP005-17; PhL-APP031) thalli of *Sargassum muticum*, collected in Venice Lagoon (Italy) (45°25′42.6″ N–12°19′50.7″ E), was soaked in 100 mL of deionized water, and the dispersion was then placed in an HTC reactor (4540 series Parr Instrument Company, IL, USA). The hydrothermal treatment was carried out under autogenous pressure and nitrogen atmosphere at three different reaction temperatures (180, 240, and 300 °C), which were monitored by a thermocouple fixed in the autoclave and connected to the reactor control. The residence time after reaching the reaction temperature was set to 60 min at a stirring speed of 300 rpm. The HTC liquid phase was separated from the solid phase by filtration under vacuum using a filter paper and a Buchner funnel. The obtained hydrochar was kept for further applications, while the corresponding liquid phases, namely AHL_180_, AHL_240_, and AHL_300_, were used for subsequent experiments.

### 4.2. Seed Priming and Germination

All experiments were performed on plants of *Phaseolus vulgaris* L. cv. ‘Borlotto’ (Adamo, Modica, Italy), one of the most common commercial cultivars in Italy [62]. Seeds were soaked for 12 h in water (control plants) or in three different AHL solutions (i.e., AHL_180_, AHL_240_, and AHL_300_; see above) [63]. To exclude morpho-physiological effects driven by a mere osmotic influence of the checked AHL solutions, seed-priming treatments using PEG 6000 solutions at the same osmotic pressure as the AHL solutions were also investigated. The osmotic potentials of the three AHL solutions were estimated by a dew point hygrometer (WP4, Decagon Devices Inc., Pullman, WA, USA) and were −0.51 MPa, −0.23 MPa, and −0.33 MPa, respectively. In summary, 7 different sets of ‘Borlotto’ seeds (45 seeds per set) were soaked in 7 different solutions: water (control samples, C), AHL_180_ solution and the corresponding PEG 6000 isotonic solution (AHL_180_ and PEG_AHL180_ samples, respectively), AHL_240_ solution and the corresponding PEG 6000 isotonic solution (AHL_240_ and PEG_AHL240_ samples, respectively), and AHL_300_ solution and the corresponding PEG 6000 isotonic solution (AHL_300_ and PEG_AHL300_ samples, respectively) (Figure 1). 

After soaking, the samples were put on a wet filter paper into different containers (5 seeds in each container, i.e., three replicas for each treatment) and incubated at 25 °C. After 7 days, germinated seeds were counted (i.e., a seed was considered germinated when the radical pierced the coats up to 2 mm), transferred in 3.4 L pots filled with forest topsoil collected from Colli San Rizzo (Messina, Italy), and regularly irrigated (twice a week) at field capacity during the whole experiment. Plants were grown in a greenhouse of the University of Messina (Italy) provided by lamps (Spectrum Grow led 6, Mastergrower, China). The photosynthetic active radiation (PAR) was 600 mmol m^−2^ s^−1^. A photoperiod of 12 h was imposed. The day/night temperature was 18/20 ± 2 °C, and the mean relative humidity was 52 ± 3%. 

Growth and water relations have been monitored up to the development of flowers and some fruits.

### 4.3. Plant Growth and Anatomical Traits

The height, number of leaves, and basal stem diameter were monitored once a week during the whole experimental period (nine weeks) in five samples per treatment. 

Eight leaves from eight different plants for each treatment were measured to estimate the leaf dry matter content (LDMC = leaf dry weight/leaf turgid weight) and the leaf mass area (LMA = leaf dry weight/leaf surface area, A_L_) [61]. In detail, A_L_ was measured by acquiring leaf images with a scanner (HP Scanjet G4050, USA) and using Image J software for the estimation. Leaves were sampled before turning on the lamps on eight-week-old well-watered plants, and their fresh weight was considered as the turgid weight.

Stomatal density was estimated on eight-week-old samples, too. Nail polish was applied on the abaxial side of three well-developed leaves per treatment and allowed to dry. Then, it was peeled off by using clear packing tape, the obtained sample was mounted on a microscope slide, and images were captured by a digital camera (Laborlux S, Leitz GmbH, Stuttgart, Germany) [64]. The number and length of stomata were estimated using Image J software.

At the end of the experimental period (i.e., on nine-week-old samples), all leaves were cut, and leaf images were acquired with a scanner in order to estimate the whole leaf surface area (A_L tot_) of each sample, as above described. Then, the plants were pulled out of the pot. The soil was gently rinsed under water to avoid damage to the root system, and the root, stem, and leaves were oven-dried for three days at 70 °C to obtain their dry weight (DW). Leaf samples were further utilized to estimate their mineral content (see below).

### 4.4. Water Relations Parameters

Maximum leaf stomatal conductance to water vapor (g_L_), transpiration rate (E_L_), and photosynthesis rate (A_n_) were measured at midday on the leaves of three plants per experimental group using a portable LCi Analyzer System (ADC Bioscientific Ltd., Herts, UK). The water use efficiency (WUE) of each measured plant was estimated by the ratio A_n_/E_L_. 

In order to quantify the differences of water relation parameters in terms of leaf water potential at the turgor loss point (Ψ_tlp_) and osmotic potential at full turgor (π_o_), they were measured on five leaves from five individuals according to Petruzzellis et al. [65]. In detail, leaves collected in the morning (before turning on the lamps) on well-watered samples were wrapped in cling film and immersed in liquid nitrogen for 2 min. Samples were then ground (still sealed in cling film) and stored in sealed plastic bottles at −20 °C until measurements. Leaf samples were then thawed at room temperature and measured by a dew point hygrometer (WP4, Decagon Devices Inc.). According to Petruzzellis et al. [65], values of π_0_ were then estimated by the following formula:π_0_ = (0.5303 × π_WP4_) + (0.0019 × LDMC) 
where π_WP4_ is the osmotic potential estimated by a dew point hygrometer, and LDMC is the leaf dry matter content (see above).

Values of Ψ_tlp_ were finally calculated by the following formula:Ψ_tlp_ = (1.31 × π_0_) − 0.03

### 4.5. Leaf Mineral Contents

Leaf samples were dried at 80 °C, cooled, and weighed in order to estimate the N and C content. After grinding, the element content was determined on randomly drawn subsamples. Total nitrogen was analyzed using a semi-micro-Kjeldahl procedure after digesting samples with H_2_SO_4_ and Kjeltab catalyst (Fisher Chemicals, Hampton, NH, USA) at 400 °C. The C/N ratio was calculated from N and C percentage data (g 100 g^−1^ dry weight).

Moreover, in order to check possible differences in the leaf nutrient content, milled samples were digested with a mixture of HNO_3_ and HClO_4_ at 200 °C with reflux cooling and analyzed by SEM-EDX (scanning electron microscopy and energy-dispersive X-ray spectroscopy) using a Zeiss 1540XB FE SEM (Zeiss, Jena, Germany) instrument operating at 10 kV [66].

### 4.6. Statistical Analysis

Data were analyzed with SigmaStat 12.0 (SPSS, Inc., Chicago, IL, USA) statistics package. To test the differences among checked solutions on all measured parameters, one-way ANOVA and post hoc Tukey tests were performed. The significance of correlations was tested using the Pearson product-moment coefficient. All differences were considered significant at *p* < 0.05.

## 5. Conclusions

Overall, our findings suggest a promising use of AHL solutions in agriculture in the framework of resource management and green economy. Indeed, they improve plant performance under well-watered conditions by increasing the photosynthesis rate and, as a consequence, shoot and/or root biomass. Last but not least, the biomass conversion of seaweeds by HTC and the reuse of AHL in agriculture are excellent examples of circular economy because they lead to (i) an environmentally sustainable solution to HCT-derived waste disposable, (ii) an incentive to use HTC in practical algal biomass management, (iii) the exploitation of algal biomass produced in eutrophic environments, and (iv) the mitigation of climate-change-driven loss in crop yields. 

Further studies need to be performed to investigate the effect of these solutions in plants growing under conditions of environmental stress, to explore their possible use to trigger plant stress resistance, and to achieve high productivity under different environmental constraints.

## Figures and Tables

**Figure 1 plants-12-02745-f001:**
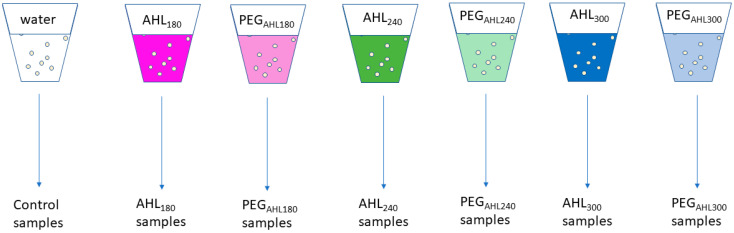
Seed-priming treatments applied in the present study. Seeds of *Phaseolus vulgaris* cv. ‘Borlotto’ were soaked in 7 different solutions: water (control samples, C), AHL_180_ solution and corresponding PEG isotonic solution (AHL_180_ and PEG_AHL180_ samples, respectively), AHL_240_ solution and corresponding PEG isotonic solution (AHL_240_ and PEG_AHL240_ samples, respectively), and AHL_300_ solution and corresponding PEG isotonic solution (AHL_300_ and PEG_AHL300_ samples, respectively). For details, see the text.

**Figure 2 plants-12-02745-f002:**
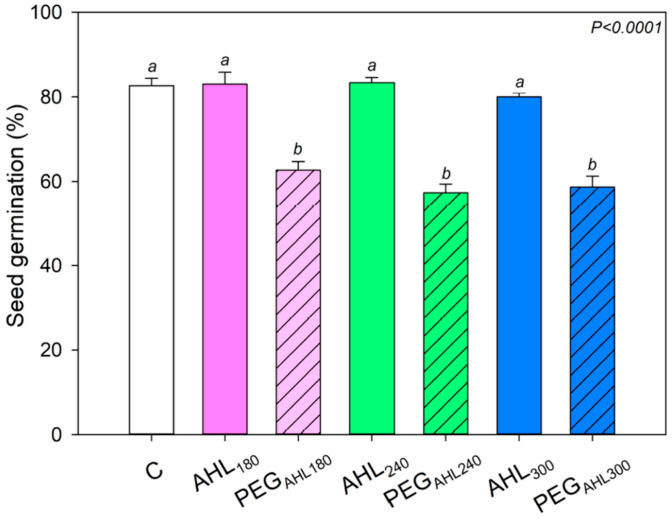
Mean value ± SD (*n* = 3) of the seed germination percentage after seed soaking into water (control samples, C), AHL_180_ solution and the corresponding PEG isotonic solution (AHL_180_ and PEG_AHL180_ samples, respectively), AHL_240_ solution and corresponding PEG isotonic solution (AHL_240_ and PEG_AHL240_ samples, respectively), and AHL_300_ solution and corresponding PEG isotonic solution (AHL_300_ and PEG_AHL300_ samples, respectively). Different letters indicate statistically different values based on one-way ANOVA test. *p* value is reported.

**Figure 3 plants-12-02745-f003:**
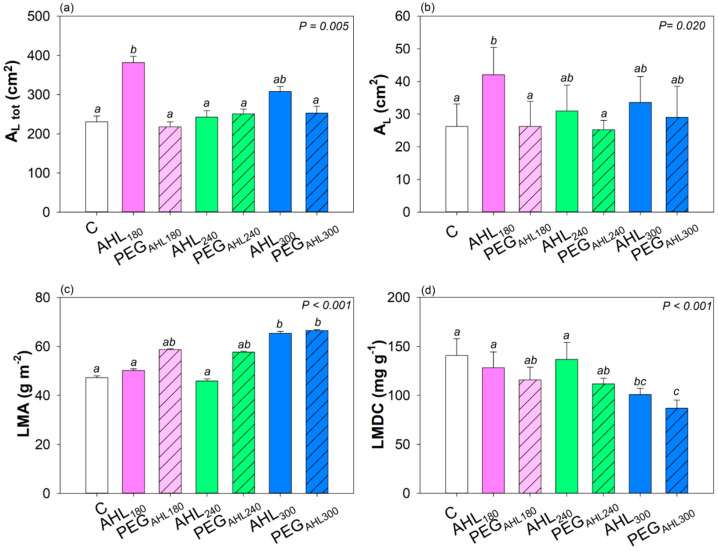
Mean value ± SD (*n* = 5) of (**a**) the whole leaf surface area (A_L tot_), (**b**) mean leaf surface area (A_L_), (**c**) leaf mass area (LMA), and (**d**) leaf dry matter content (LDMC) as recorded in samples of *Phaseolus vulgaris* cv. ‘Borlotto’ in which seeds were soaked into water (control samples, C), AHL_180_ solution and the corresponding PEG isotonic solution (AHL_180_ and PEG_AHL180_ samples, respectively), AHL_240_ solution and corresponding PEG isotonic solution (AHL_240_ and PEG_AHL240_ samples, respectively), and AHL_300_ solution and corresponding PEG isotonic solution (AHL_300_ and PEG_AHL300_ samples, respectively). Different letters indicate statistically different values based on one-way ANOVA test. *p* values are reported.

**Figure 4 plants-12-02745-f004:**
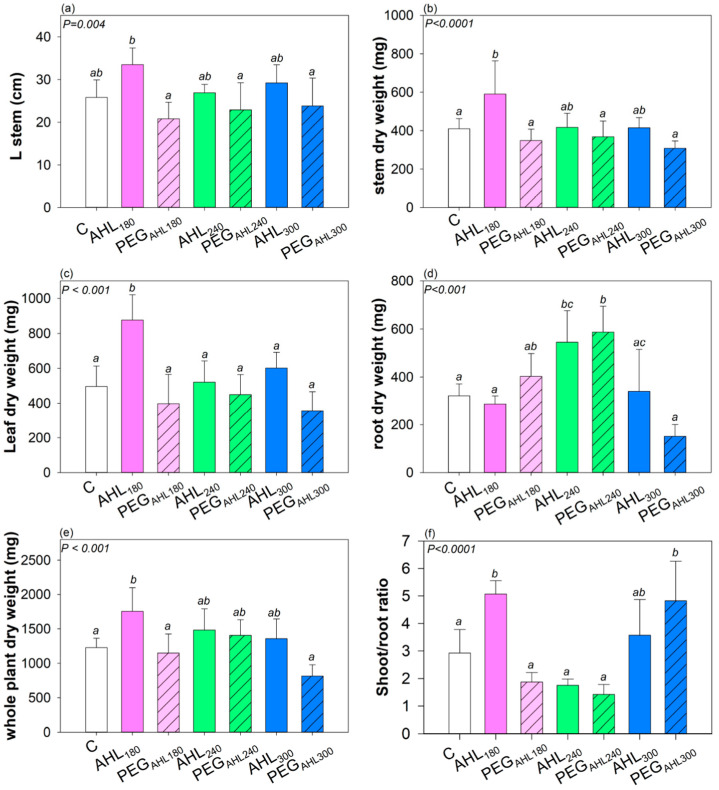
Mean value ± SD (*n* = 5) of (**a**) the stem length (L stem), (**b**) stem dry weight, (**c**) leaf dry weight, (**d**) root dry weight, (**e**) whole plant dry weight, and (**f**) shoot/root ratio, as recorded in samples in which seeds were soaked into water (control samples, C), AHL_180_ solution and the corresponding PEG isotonic solution (AHL_180_ and PEG_AHL180_ samples, respectively), AHL_240_ solution and corresponding PEG isotonic solution (AHL_240_ and PEG_AHL240_ samples, respectively), and AHL_300_ solution and corresponding PEG isotonic solution (AHL_300_ and PEG_AHL300_ samples, respectively). Different letters indicate statistically different values based on one-way ANOVA test. *p* values are reported.

**Figure 5 plants-12-02745-f005:**
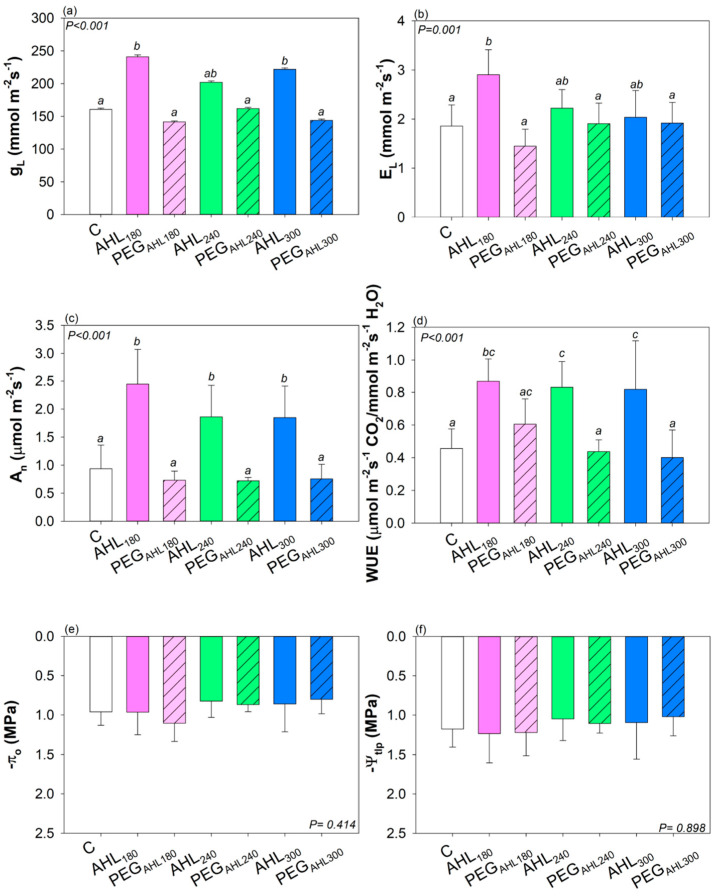
Mean value ± SD (*n* = 5) of (**a**) stomatal conductance to water vapor (g_L_), (**b**) transpiration rate (E_L_), (**c**) photosynthesis rate (A_n_), (**d**) water use efficiency (WUE), (**e**) osmotic potential at full turgor (π_o_), and (**f**) leaf water potential at turgor loss point (Ψ_tlp_) as recorded in samples in which seed were soaked into water (control samples, C), AHL_180_ solution and the corresponding PEG isotonic solution (AHL_180_ and PEG_AHL180_ samples, respectively), AHL_240_ solution and corresponding PEG isotonic solution (AHL_240_ and PEG_AHL240_ samples, respectively), and AHL_300_ solution and corresponding PEG isotonic solution (AHL_300_ and PEG_AHL300_ samples, respectively). Different letters indicate statistically different values based on one-way ANOVA test. *p* values are reported.

**Figure 6 plants-12-02745-f006:**
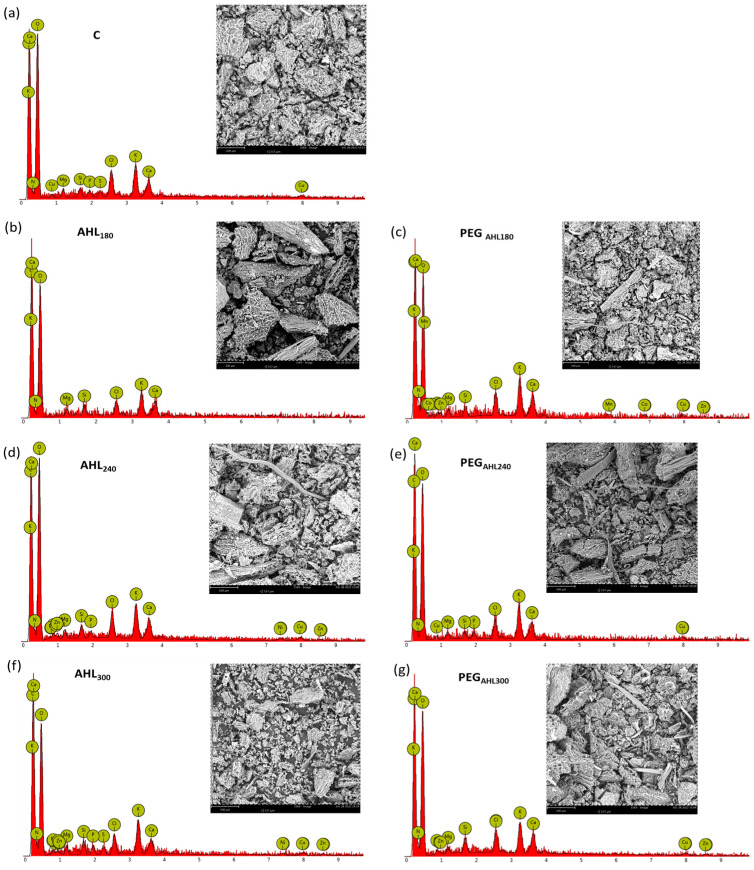
Scanning electron microscope images energy-dispersive X-ray spectra of dry and grinded leaf samples as recorded in samples in which seeds were soaked into (**a**) water (control samples, C), (**b**) AHL_180_ solution and the (**c**) corresponding PEG isotonic solution (AHL_180_ and PEG_AHL180_ samples, respectively), (**d**) AHL_240_ solution and (**e**) corresponding PEG isotonic solution (AHL_240_ and PEG_AHL240_ samples, respectively), and (**f**) AHL_300_ solution and (**g**) corresponding PEG isotonic solution (AHL_300_ and PEG_AHL300_ samples, respectively). Scale bars correspond to 100 µm.

**Table 1 plants-12-02745-t001:** Mean values ± SD of stomatal density and values of carbon content, nitrogen content, and carbon–nitrogen ratio recorded in leaf samples of *Phaseolus vulgaris* cv. ‘Borlotto’. C: control samples; AHL_180_ and PEG_AHL180_: seeds treated with AHL_180_ solution and corresponding PEG isotonic solution, respectively; AHL_240_ and PEG_AHL240_: samples treated with AHL_240_ solution and corresponding PEG isotonic solution, respectively; AHL_300_ and PEG_AHL300_: samples treated with AHL_300_ solution and corresponding PEG isotonic solution, respectively. Different letters indicate statistically different values based on one-way ANOVA test (*p* < 0.05).

	Stomatal Density (m^−2^)	C (mg g^−1^)	C (%)	N (mg g^−1^)	N (%)	C:N
C	148.4 ± 9.9 a	76.83	41.83	5.141	2.726	15.3
AHL_180_	176.7 ± 10.2 ab	73.80	41.93	5.644	3.124	13.4
PEG_AHL180_	131.0 ± 28.5 ac	78.94	41.96	5.262	2.725	15.4
AHL_240_	182.6 ± 8.4 ab	72.49	41.74	5.272	2.956	14.1
PEG_AHL240_	104.7 ± 8.3 c	67.64	41.62	5.8	3.476	12
AHL_300_	188.2 ± 9.5 b	69.41	41.73	6.544	3.833	10.9
PEG_AHL300_	148.7 ± 2.5 a	77.85	41.7	7.167	3.74	11.2

## Data Availability

The data presented in this study are available on request from the corresponding author.

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
