# Peer review of "Using the Aqueous Phase Produced from Hydrothermal Carbonization Process of Brown Seaweed to Improve the Growth of Phaseolus vulgaris"

_plants, 2023, doi:10.3390/plants12142745_

Round 1
Reviewer 1 Report
The article initiated an interesting method to improve the growth of Phaseolus vulgaris. The paper revealed the impact of seed priming treatment with HTC-discarded liquid phases on bean seed germination, plant growth and productivity grown in a greenhouse. The result would be important in making exact AHL solutions reuse in agriculture. In my opinion, the results are relatively reliable, the figures and content were well organized, and some revisions need to be done before the paper can be accepted.
1. In the Results section, the trends of the observed data between different treatments are well expressed, the authors should add some quantitative or comparative data to some of the key results of the paper.
2. Line 131, “root/shoot ratio” should be replaced by “shoot/root ratio”.
3. Figure 6 is too small to distinguish the ionic components.
4. Line 249-250, It is suggested to give the main nutritional components of three AHL solutions.
5. Line 345-349, please indicate the source or reference of these two formulas.
Author Response
Reviewer 1
The article initiated an interesting method to improve the growth of Phaseolus vulgaris. The paper revealed the impact of seed priming treatment with HTC-discarded liquid phases on bean seed germination, plant growth and productivity grown in a greenhouse. The result would be important in making exact AHL solutions reuse in agriculture. In my opinion, the results are relatively reliable, the figures and content were well organized, and some revisions need to be done before the paper can be accepted.
Thank you for the overall positive comment on our work.
- In the Results section, the trends of the observed data between different treatments are well expressed, the authors should add some quantitative or comparative data to some of the key results of the paper.
Thank you, nevertheless, we prefer to discuss data in the Discussion.
- Line 131, “root/shoot ratio” should be replaced by “shoot/root ratio”.
Done.
- Figure 6 is too small to distinguish the ionic components.
Thank you for the suggestion, we edited the Figure.
- Lines 249-250, It is suggested to give the main nutritional components of three AHL solutions.
As reported in the Introduction, the aim of this study was to i) investigate the potential benefit(s) of a seed priming treatment using AHL solution and ii) disentangle the potential benefit(s) of this treatment from that of a mere osmotic effect (see lines 80-84). For this reason, we didn’t investigate the chemical composition of the AHL solutions in this study. The knowledge of the beneficial effects of some discarded solutions obtained with a specific HTC protocol (i.e., AHL180) is the starting point for future studies, which will also investigate the chemical composition of these solutions.
- Lines 345-349, please indicate the source or reference of these two formulas.
Thank you for the suggestion. The reference was added (line 344).
Reviewer 2 Report
Dear Authors
The manuscript titled ,,Using the aqueous phase produced from hydrothermal carbonization process of brown seaweed to improve the growth of Phaseolus vulgaris’’ is written clearly and concisely. It has high scientific value and the conclusions reached by the authors are satisfying. However, I have some comments that should be taken into account by the Authors when revising this manuscript.
1. Please provide the research hypothesis at the end of the Introduction section. This research hypothesis should be referred to in the Conclusions chapter.
2. The description of seed priming methods (line 83-94 in the Introduction) should be moved to the Materials and Methods section.
3. Figures 3-4 captions - delete the sentence ‘’ For details on seed treatment, see the text’’.
4. Materials and Methods section –
A) Explain why cv. ‘Borlotto’ was chosen for testing
B) Provide references to the methods described in the subsections: ,, 4.2 Seed priming and germination’’; ,,4.3 Plant growth and anatomical traits’’; ,, 4.4 Water relations parameters’’; ,, 4.5 Leaf mineral contents’’
5. The Conclusions chapter needs a major rewrite. References should not be included in this chapter. The main part of the conclusions should refer to the research hypothesis and the results of your own research.
Author Response
Reviewer 2
The manuscript titled “Using the aqueous phase produced from hydrothermal carbonization process of brown seaweed to improve the growth of Phaseolus vulgaris’’ is written clearly and concisely. It has high scientific value and the conclusions reached by the authors are satisfying. However, I have some comments that should be taken into account by the Authors when revising this manuscript.
Thank you for the positive comments.
- Please provide the research hypothesis at the end of the Introduction section. This research hypothesis should be referred to in the Conclusions chapter.
Thank you for this comment. Expected results has been now inserted in the revised manuscript (L. 85-87).
- The description of seed priming methods (line 83-94 in the Introduction) should be moved to the Materials and Methods section.
Figure 1 describes the experimental treatment and, therefore, was moved to Material and Methods. However, based on the Instruction to Authors, Material and Methods is at the end of the manuscript. Therefore, we believe that a first explanation on the experimental planning, can help the reader to read the results and discussion.
- Figures 3-4 captions - delete the sentence ‘’ For details on seed treatment, see the text’’.
Done.
- Materials and Methods section –
- A) Explain why cv. ‘Borlotto’ was chosen for testing
The Borlotto is one of the most used bean cultivars in the world. This explanation was added in the text (line 285).
- B) Provide references to the methods described in the subsections: ,, 4.2 Seed priming and germination’’; ,,4.3 Plant growth and anatomical traits’’; ,, 4.4 Water relations parameters’’; ,, 4.5 Leaf mineral contents’’
Done (lines 287, 315, 344, 361).
- The Conclusions chapter needs a major rewrite. References should not be included in this chapter. The main part of the conclusions should refer to the research hypothesis and the results of your own research.
Thank you for the suggestion. This chapter was revised.
Round 2
Reviewer 1 Report
The authors have made a good revision, and I recommend accepting the manuscript for publication.
Author Response
Thank you
Reviewer 2 Report
Dear Authors
The manuscript titled ,,Using the aqueous phase produced from hydrothermal carbonization process of brown seaweed to improve the growth of Phaseolus vulgaris’’ needs further revision.
1. The description of seed priming methods should be moved to the Materials and Methods section. - it has not been corrected.
2. Materials and Methods section –
A) Explain why cv. ‘Borlotto’ was chosen for testing - it is stated that "cv. ‘Borlotto’ is one of the most used bean cultivars in the world. References to this information are needed as well as the morphological characteristics and agricultural value of this cultivar.
3. The Conclusions chapter still needs a major rewrite. There are no conclusions regarding the parameters tested in the experiment (plant growth and anatomical traits).
Author Response
The manuscript titled “Using the aqueous phase produced from hydrothermal carbonization process of brown seaweed to improve the growth of Phaseolus vulgaris” needs further revision.
- The description of seed priming methods should be moved to the Materials and Methods section. - it has not been corrected.
Thanks for your suggestion, however, in our opinion, it is relevant to introduce the seed priming treatment in the Introduction, giving references on this technique, in order to justify the aims of our research plan, as done for HTC technique.
- Materials and Methods section –
- A) Explain why cv. ‘Borlotto’ was chosen for testing - it is stated that "cv. ‘Borlotto’ is one of the most used bean cultivars in the world. References to this information are needed as well as the morphological characteristics and agricultural value of this cultivar.
Done (L. 290-291).
- The Conclusions chapter still needs a major rewrite. There are no conclusions regarding the parameters tested in the experiment (plant growth and anatomical traits).
Done.